# Biologically-Inspired Spatial Neural Networks

**Maciej Wołczyk**
Jagiellonian University
maciej.wolczyk@gmail.com

**Jacek Tabor**
Jagiellonian University
jacek.tabor@uj.edu.pl

**Marek Śmieja**
Jagiellonian University
marek.smieja@uj.edu.pl

**Szymon Maszke**
Jagiellonian University
szymon.maszke@gmail.com

## Abstract

We introduce bio-inspired artificial neural networks consisting of neurons that are additionally characterized by spatial positions. To simulate properties of biological systems we add the costs penalizing long connections and the proximity of neurons in a two-dimensional space. Our experiments show that in the case where the network performs two different tasks, the neurons naturally split into clusters, where each cluster is responsible for processing a different task. This behavior not only corresponds to the biological systems, but also allows for further insight into interpretability and continual learning.

## 1 Introduction

Neurons in the human brain naturally group into interconnected regions, forming the full neural system [1]. In this paper, we would like to construct an analogical mechanism in the case of artificial neural networks. To put this idea into practice, we supply each neuron with spatial coordinates. Motivated by biological neural systems, we impose the cost of signal transmission between connected neurons, which grows linearly with the distance between them. In consequence, we obtain artificial groups specialized in different tasks, each group containing neurons that are placed close to each other.

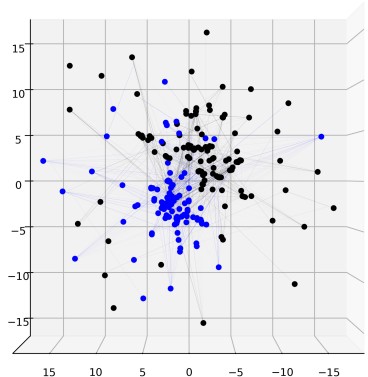

Figure 1: Top-down view of all neurons in the proposed spatial network. The network naturally forms two clusters of neurons corresponding to each task, represented by different colors.

The proposed model is examined in a double classification task, where a single network has to classify examples from two different datasets (MNIST and Fashion-MNIST). At test time, we split the network into two subnetworks based on the structure of weights, where each subnetwork represents one task. The resulting models perform their respective tasks only slightly worse than the original network, in contrast to the large performance drop observed after splitting a standard fully connected network. Our model offers a natural interpretation of neurons' responsibilities and is analogous to biological neural systems.

The idea of adding spatial coordinates to each neuron and penalizing long connections was previously introduced by [2]. Although our work is based on a similar premise, the resulting models differ significantly. We use a different, simpler spatial loss

function, we investigate networks with multiple hidden layers as opposed to a single hidden layer, and we focus on the cluster-forming properties of spatial networks.

There have been multiple approaches to finding clusters of neurons in artificial neural networks, although most of them consider the functional aspects of the network. For instance, [3] compare variance of neuron's activations in different tasks to decide which cluster does it belong to. In another approach, [4] cluster the network by using feature vectors calculated based on correlation between the neuron and the output. In comparison, our approach uses the structure of the network – the spatial placement of the neurons and the strength of connections between the neurons.

Our model is also related to continual and multi-task learning [5] and parameter reduction models for deep neural networks [6]. In particular, the effect is slightly similar to the one obtained in [7]. Authors focus on splitting network weights into a set of groups, where each is associated with a class (task). In contrast to our biologically inspired mechanism, [7] use a specialized weight regularization technique and strive towards a different goal. In [8], a nested sparse network is constructed and different forms of knowledge are learned at each level, enabling solving multiple tasks with a single neural network. Checking the response of different groups in our proposed spatial network could be considered useful for interpreting the network predictions, which is also an open problem [9].

## 2 Spatial neural network

In this section, we introduce our spatial network model and describe its components.

**Setting.** Let us consider a typical fully-connected neural network. Each neuron $n$ is then additionally characterized by two-dimensional spatial features $p(n) \in \mathbb{R}^2$, which describe its position in the space of the layer it belongs to. The positions are parameters of the model (i.e. we update them during gradient descent iterations) and the initial coordinates are sampled from a standard Gaussian distribution. To encourage movement and grouping of the neurons, we introduce additional loss components to the final cost function of the model.

**Transport cost.** In biological systems transferring information over a long distance is not energetically efficient and may cause unnecessary delays [10]. Inspired by this fact, we want to penalize strong connections between neurons that are far away from each other. Thus, in a given layer $l$ we use the spatial L1 penalty, which takes into account the distance between the neurons:

$$T(l) = \tfrac{1}{N_l} \sum_{n_1 \to n_2, n_2 \in l} |a_{n_1 n_2}| \cdot \|p(n_1) - p(n_2)\|.$$

where $N_l$ is the number of neurons in the layer $l$, $n_1 \to n_2$ denotes that we consider the connection from the neuron $n_1$ to $n_2$, $a_{n_1 n_2}$ denotes the weight of the connection and $n_2 \in l$ means that the neuron $n_2$ belongs to the layer $l$.

**Density of neurons.** On the other hand, the neurons should not be packed too densely. To avoid this we add, as it is common, a loss component based on the negative force between neurons (defined by a potential). The density cost for the layer $l$ is then defined as:

$$V(l) = \tfrac{1}{N_l^2} \sum_{n_1, n_2 \in l} \exp(-\|p(n_1) - p(n_2)\|),$$

where $N_l$ is the number of neurons in the layer $l$.

We apply those costs only to the set $\mathcal{L}$ of selected layers to allow the network more flexibility. The final loss function of our spatial network model is then:

$$L_{\text{spatial}} = L + \tfrac{1}{|\mathcal{L}|} \sum_{l \in \mathcal{L}} \alpha T(l) + \beta V(l),$$

where $L$ is the original loss function of the model, and $\alpha$ and $\beta$ are hyperparameters used for weighting out the components of the loss function.

Table 1: Averaged accuracy on test sets before and after splitting the network into independent groups of neurons. We compare our spatial network to the regular network for three different input representations. Each experiment was run three times – means and standard deviations are provided.

| | Concatenation | | Mixing | | Sequential | |
|---|---|---|---|---|---|---|
| | Regular | Spatial | Regular | Spatial | Regular | Spatial |
| Full network | $0.92 \pm 0.04$ | $0.92 \pm 0.04$ | $0.88 \pm 0.04$ | $0.89 \pm 0.04$ | $0.92 \pm 0.05$ | $0.92 \pm 0.05$ |
| Split network | $0.80 \pm 0.13$ | $0.92 \pm 0.04$ | $0.70 \pm 0.17$ | $0.89 \pm 0.03$ | $0.33 \pm 0.20$ | $0.84 \pm 0.08$ |
| Acc. drop | $0.11 \pm 0.15$ | $0.00 \pm 0.00$ | $0.19 \pm 0.20$ | $0.00 \pm 0.00$ | $0.60 \pm 0.20$ | $0.07 \pm 0.07$ |

**Activation function.** We have chosen to use the sigmoidal activation function. This is in part because of its connection to the biological systems, but also because of the rescaling invariance of the most commonly used ReLU. In the case of the said activation function, we can jointly rescale the weights without changing the final result of the network. Consequently, the network could easily minimize the loss function without changing its spatial structure. This hypothesis seemed to be confirmed by our preliminary experiments.

# 3 Experiments

We measure how the spatial network performs on a double classification task – i.e. the network has to simultaneously classify examples belonging to two different datasets, MNIST and Fashion-MNIST. We have chosen those due to similarities in their structure (the same number of classes and input dimensions) and different semantics (i.e. disjoint classes). Since the classification tasks are mostly independent, we hypothesize that the network should be able to split its neurons into groups representing each task. The code is available at `https://github.com/gmum/SpatialNetworks`.

**Implementation details.** Our method was tested with three different ways of passing the inputs to the network:

1. **Concatenated inputs** – input is a vector of dimension 1568 obtained by concatenating flattened examples from MNIST and Fashion-MNIST (one from each dataset).

2. **Mixed inputs** – input is a vector of dimension 784, which is a linear combination of a MNIST and a Fashion-MNIST example (i.e. element-wise addition).

3. **Sequential inputs** – input is a single example randomly sampled from both datasets.

In all three cases, the network outputs a 20-dimensional vector, which is then divided into two 10-dimensional vectors. The softmax function is then separately applied to both vectors to obtain class probabilities for each task.

The same network architecture is used for all tasks: $n - 128 - 128 - 128 - 256 - 20$, where $n$ represents the dimension of input vector and integers are the numbers of neurons in consecutive layers. We apply the spatial penalties starting from the third hidden layer and we also only split those layers. Hyperparameters are kept the same for all tasks: $\alpha = 1, \beta = 3$, learning rate is set to 0.005, batch size is 256. As a baseline, we use an analogical network without transport and density penalties.

**Splitting the network.** Our goal is to check whether in the described setting we would be able to observe the formation of disjoint regions, similar to those forming in the brain [11]. To do so, we split the network into two subnetworks responsible for different tasks, based on the structure of the connections. Then we evaluate each subnetwork (i.e. we disable the neurons that were not assigned to it) and compare its performance on its respective task to the full network.

We have chosen a simple greedy method of splitting the network. Since we use separate outputs for each task, the assignment of neurons in the last layer is given and we can proceed recursively by propagating the split backward. Then, we assign each neuron to the task it has the strongest connection to, which we measure as the sum of absolute weights of connections going from that neuron to the neurons in the next layer responsible for the given task. In other words, if $M$ is the

number of desired regions, then the neuron assignment $g(n) \in \{1, \ldots, M\}$ is given by:

$$g(n) = \arg\max_{i=1,\ldots,M} \sum_{n \to m, g(m)=i} |a_{nm}|.$$

In our case $M = 2$ for all experiments. Intuitively, if the network's neurons are in fact forming two disjoint clusters representing each task, then the inter-task connections should be close to zero, and thus the neuron will be assigned to the corresponding group[1].

**Results.** Performance of models in terms of classification accuracy before and after the split is presented in Table 1 (we only split the layers from the set $\mathcal{L}$ for which the spatial losses were applied). The results show that the spatial network can divide the neurons into task-specific subgroups in a more efficient manner than the regular network[2]. We suspect that this is because the network places neurons representing different tasks into separate clusters, which would make the task of dividing the network straightforward.

Visual inspection of the neurons' positions seems to confirm this assumption. Figure 2 shows a scatter plot of the neurons in the split layers and their group assignments obtained by our greedy method[3]. Neurons in the output layer representing different tasks are clearly separated, with groups in the previous layers mirroring this arrangement. The top-down view of the same network presented in Figure 1 shows that the neurons are indeed mostly divided into two groups with a number of outsiders around. This behavior resembles the region-forming processes in the brain that we wanted to mimic.

The method of presenting the input strongly affects the results. Even the regular network is able to perform well after the split when using the concatenated input – presumably because this imposes a structural prior encouraging the network to form two separate parts. On the other hand, the sequential task is the most difficult – we hypothesize that since the network does not perform both tasks at the same time, it is more inclined to use neurons for both tasks, which leads to a decrease in performance after the split.

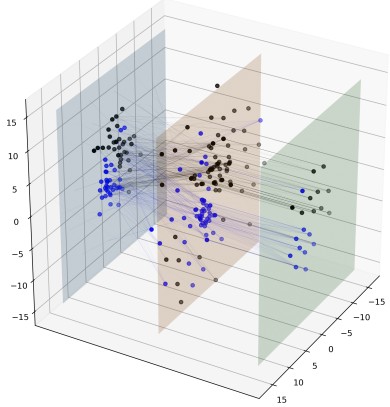

Figure 2: Three-dimensional perspective of the network shown in Figure 1. Each slice represents a different layer of the network, the rightmost being the output layer.

## 4   Conclusion

We have presented a connection between neuroscience and machine learning, which to our best knowledge has not yet been explored. Experiments show that our proposed spatial artificial neural network manifests behavior similar to the region-forming processes in the brain.

For future work we plan to test our model on a continual learning task. We hypothesize that, since the model is able to create disjoint clusters of neurons responsible for different tasks, learning a new task could be possible without disturbing the previously created clusters. Additional constraints could be added to achieve this, such as restricting the movement of neurons with high potential, while allowing neurons with low potential to move freely.

Spatial networks could also be investigated in relation to spiking neural networks. The motivation is that spiking neural networks operate in the temporal dimension, which in the brain is dictated mainly

---

[1]We have also tried splitting based on spectral clustering and using network activation statistics, but those methods failed completely, e.g. all outputs were assigned to the same task.

[2]It is also worth noting that the additional loss functions added to the spatial network do not negatively impact the performance of the model before the split.

[3]For readability, we only show neurons in last three layers (i.e. those that were split) for which the sum of absolute input weights was higher than $0.1$ and connections of absolute weight larger than $0.01$

by the spatial structure of neurons. Thus, it is possible that the two types of networks have similar properties, making the spatial network a more interesting model to investigate from the neuroscientific standpoint.

## Acknowledgements

We thank Klaudia Bałazy for help with preparing the camera-ready version of this paper.

The first author carried out this work within the research project "Bio-inspired artificial neural networks" (grant no. POIR.04.04.00-00-14DE/18-00) within the Team-Net program of the Foundation for Polish Science co-financed by the European Union under the European Regional Development Fund.

The work of the second was supported by the National Science Centre (Poland) grant no. 2017/25/B/ST6/01271.

The work of the third author was supported by the grant no. LIDER/37/0137/L-9/17/NCBR/2018 from the Polish National Centre for Research and Development.

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
