# OpenReview forum: "Biologically-Inspired Spatial Neural Networks"
_NeurIPS.cc/2019/Workshop/Neuro_AI — Real Neurons & Hidden Units @ NeurIPS 2019 Poster_

### Official Review · AnonReviewer3 · 2019-09-26
**A simple network model that forms spatially clustered regions like the brain**

**Clarity:** 4

**Comment:**

This paper seems like a first, most basic "sanity check" that could be done to try to explain in models why the brain forms computationally distinct regions that are also separated in space. While ultimately they will want to take on the more ambitious goal of making the case that their proposed mechanism is truly the primary reason this happens, the scope of the work seems appropriate for a workshop.

I think the work would have benefitted from a measure of how close the connections decompose into non-overlapping subgraphs, without taking into consideration the labels, and to show that these "anatomical" subgraphs are the same found through their strategy used to find "functional" subgraphs via backpropagating label assignments to the neurons. This could help answer the question of if the connectome is sufficient information for defining regions in the brain. As stated above, more discussion as to the "interpretation" of their subgraph decomposition method, and comparisons with other possible ways to do this, would have been helpful.

Some discussion of recurrent connections in a paper meant to model the brain would have been beneficial.

**Category:**

AI->Neuro

**Clarity Comment:**

The technical aspects of the paper and the basic reasoning behind them are clear. However, I feel that the motivation behind the work and some of the technical choices that were made could have been clarified somewhat. More discussion as to the "interpretation" of their subgraph decomposition method, and comparisons with other possible ways to do this, would have been helpful.

Since the results are largely what one would expect to see, it would have given the work's purpose more clarity if they had suggested future steps that could push the work further, or provided some sort of ultimate "end goal" for this line of inquiry.

**Evaluation:**

3: Good

**Importance:**

3: Important

**Importance Comment:**

The authors shed light on how/why spatially distinct regions with different computational roles form in the brain. The authors show the interesting although not too surprising result that penalizing long-range connections results in networks that are functionally and to some extent topologically compartmentalized into spatially separated subgroups of neurons. The impact is diminished by the work lacking a roadmap to further inquiry.

**Intersection:**

3: Medium

**Intersection Comment:**

The paper as written is concerned with using artificial neural networks to help explain biological ones, without a clear path to closing the loop by going in the other direction. As such, doesn't seem likely to be very interesting to an AI researcher as written.

**Rigor Comment:**

The mechanisms they use (primarily the l1 penalty on neural distances) are clearly described, as is the method for splitting the network into two subnetworks. Comparisons with other mechanisms (such as an l2 penalty on neural distances) would have been helpful, and it would have been a worthwhile endeavor to show that their model is in some sense the most natural or minimal model that generates the desired phenomena.

The evidence provided by the authors that these subgroups are spatially isolated from each other is visual. Quantitative measures would have made the point more convincing.

The authors address the issue of input encoding in a direct, thorough, and convincing way.

There may be a better way to assign class labels to hidden neurons than the greedy algorithm they propose. One potential issue that I see is that a given neuron in layer l may have strong connections to two neurons in layer l+1 that themselves are assigned to the same class, but the outputs of these two neurons may cancel out in layer l+2. It might be worth looking at the change of the loss with respect to changes in the hidden layer neurons in order to assign the labels. The authors may have already tried this approach -- I couldn't really tell from the footnote they wrote on the matter.

**Technical Rigor:**

4: Very convincing

---

### Official Review · AnonReviewer1 · 2019-09-26
**Interesting idea and results, and perhaps raises more questions in the process**

**Clarity:** 4

**Comment:**

This is an interesting paper that attempts to analyze learned neural representations when neurons have additional spatial properties which determine their connection lengths. The network is then trained with a loss function that accounts for the strengths of the connections and penalizes large distances. This forces the resulting neurons within one layer to cluster together during a two-task classification.

**Category:**

Neuro->AI

**Clarity Comment:**

This paper is well written. The concepts, equations, and connections are emphasized and generally understood. Figures are also simple and clearly understandable.

**Evaluation:**

3: Good

**Importance:**

3: Important

**Importance Comment:**

This is an interesting paper and perhaps generates more questions than it answers, such as connection with other local learning rules. The question being asked is certainly interesting and relevant.

**Intersection:**

4: High

**Intersection Comment:**

The authors use an MLP trained through backprop with penalty constraints to see if the learned network can be split to solve separate tasks. The biological connection is with constrained learning in the brain based on spatial constraints. The results are interesting, although it’s not immediately clear what the contribution are for neuroscience / ML.

**Rigor Comment:**

The technicalities are straightforward and justified. It would, however, be interesting to see more analyses on properties during training, including convergence compared to benchmark without the additional spatial losses. In addition, how robust is this finding relative to network architecture?

**Technical Rigor:**

4: Very convincing

---

### Official Review · AnonReviewer2 · 2019-09-27
**Good first step in studying the effects of wiring constraints on modularity of a network trained on multiple tasks - quite preliminary**

**Clarity:** 4

**Comment:**

Suggestions for next steps:
Is the modularity an effect of spatial constraint or weight strength constraint? Try both indpendently.
What if the network has limited capacity (fewer neurons per layer)? Does it share more neurons between tasks in this case?

**Category:**

AI->Neuro

**Clarity Comment:**

Clearly presented.

Please connect your work to this relevant work:
https://ieeexplore.ieee.org/document/6793887
https://royalsocietypublishing.org/doi/full/10.1098/rspb.2012.2863
https://www.cell.com/neuron/fulltext/S0896-6273(18)30250-2
https://www.nature.com/articles/s41593-018-0310-2
https://arxiv.org/abs/1909.09847 (just came out)

Minor: in the cost function,  I believe "alphaT(l)+ L(l)" should be "alphaT(l)+ V(l)".



**Evaluation:**

3: Good

**Importance:**

3: Important

**Importance Comment:**

By introducing a cost on wiring strength and length to a fully-connected feedforward neural network, the authors show that this network trained on two tasks simultaneously (MNIST and Fashion-MNIST) splits into two modular and spatially segregated networks. This result is expected and quite preliminary but nonetheless interesting and relevant to the workshop.

**Intersection:**

4: High

**Intersection Comment:**

This work could be relevant both to understand biological neural networks as well as build more efficient artificial neural networks.

**Rigor Comment:**

Rigorous

**Technical Rigor:**

4: Very convincing

---

### Decision · Program_Chairs · 2019-10-02

Accept (Poster)